# The Molten Globule State of a Globular Protein in a Cell Is More or Less Frequent Case Rather than an Exception

**DOI:** 10.3390/molecules27144361

**Published:** 2022-07-07

**Authors:** Valentina E. Bychkova, Dmitry A. Dolgikh, Vitalii A. Balobanov, Alexei V. Finkelstein

**Affiliations:** 1Institute of Protein Research, Russian Academy of Sciences, 142290 Pushchino, Moscow Region, Russia; vbychkova@mail.ru (V.E.B.); afinkel@vega.protres.ru (A.V.F.); 2Shemyakin-Ovchinnikov Institute of Bioorganic Chemistry, Russian Academy of Sciences, 117871 Moscow, Russia; dolgikh@nmr.ru

**Keywords:** globular protein, rigid native state, molten globule, intrinsically disordered, functional state, unfolded state, coil, post-translational modifications, membrane, chaperone

## Abstract

Quite a long time ago, Oleg B. Ptitsyn put forward a hypothesis about the possible functional significance of the molten globule (MG) state for the functioning of proteins. MG is an intermediate between the unfolded and the native state of a protein. Its experimental detection and investigation in a cell are extremely difficult. In the last decades, intensive studies have demonstrated that the MG-like state of some globular proteins arises from either their modifications or interactions with protein partners or other cell components. This review summarizes such reports. In many cases, MG was evidenced to be functionally important. Thus, the MG state is quite common for functional cellular proteins. This supports Ptitsyn’s hypothesis that some globular proteins may switch between two active states, rigid (N) and soft (MG), to work in solution or interact with partners.

## 1. Introduction

The compact state of a protein molecule, characterized by a pronounced secondary and fluctuating tertiary structure, was theoretically predicted by O.B. Ptitsyn [1] and experimentally proved by his team [2,3,4]. Later, this state was called a “molten globule” [5,6,7], and was in vitro observed for many proteins under moderately denaturing conditions in vitro (see reviews [8,9,10]).

The last three decades demonstrated a significant progress in the theory of protein folding [11,12,13,14,15,16,17,18,19,20,21,22] and intensive studies of a wide range of proteins [23,24,25,26,27,28,29,30,31,32,33]. The development of experimental approaches and the use of new techniques, especially such as nuclear magnetic resonance NMR (in some modifications) and fluorescence, made it possible to follow changes in the protein structure under cell-like conditions [24,25,26,27,28,29,30,31,32,33]. Many studies directly or indirectly imply the presence of molten protein globules under these conditions.

The discovery of intrinsically disordered (or natively unfolded) proteins (IDPs) comprehensively described by Tompa [34] (see reviews [35,36,37]) revealed that some of them are in the MG state, while the structure of others is closer to the unfolded state. The latter are sometimes referred to as “pre-MGs”. It has become clear that many protein functions require the rigid N state of the protein, while the others require their more or less disordered states. The latter have several properties similar to the MG state, but some properties distinguish typical IDPs from typical MGs. (see Table 1 and [38,39,40,41,42,43]).

It is possible to distinguish the MG state of a protein (or, by the original definition, “protein with fluctuating tertiary structure”) from IDP due to the differences in changes of their properties under different impacts.

Here, we do not discuss natively unfolded proteins but focus on the MG of “normal” globular proteins.

There are three important issues to be addressed:(1)Why proteins adopt the MG state;(2)What impacts cellular proteins causing a change in their stability and transition to MG;(3)What functions the molten protein globule performs in the cell.

For convenience, we divide issue (2) into subsections by the impact type:-Post-translational protein modifications: acetylation, phosphorylation, ubiquitination, glycation, and others.-Protein–protein interactions: substrate–receptor, multimers, modular proteins.-Protein–membrane interactions.-Protein–chaperone interactions.-Protein interactions with specific adapter proteins.

## 2. Physics of the MG

Depending on ambient conditions, the most stable state of a protein chain may be neither rigid (solid) nor completely unfolded (coil) but “molten”.

In the majority (but not all) of proteins, the MG in vitro arises either from the N state under the effect of a moderate denaturant, with the further transition to the coil as the denaturant concentration grows, or from the coil due to denaturant dilution [9]. The MG-like state also results from heat denaturation (melting) of a rigid globule. The N-to-MG transition is of the “all-or-none” type. Typically, MG does not undergo further “all-or-none” melting or swelling; rather, its unfolding looks like a cooperative though broad S-shaped transition observed by optical methods such as CD and fluorescence [9]. However, some rigid proteins (especially small ones) unfold directly into coils without any intermediate state [44].

MG is a “soft” [9] state that shows similarity to a rigid protein in many aspects. Reinforced by hydrogen bonds, its secondary structure is a well-developed and stable until the globule is “dissolved” by a solvent. However, the MG’s side chains lose their dense packing but acquire freedom of movement (that is, they lose energy but gain entropy). The liberation of the side chain rotational isomerization is the main driving force of protein melting [45].

Since most of the protein chain degrees of freedom relate to the small-scale side chain movements, it is their liberation that can make the MG thermodynamically advantageous. The liberation of small-scale side chain rotational isomerization does not require the complete unfolding of the globule; slight swelling would be enough. This swelling, however, leads to a significant decrease in the van der Waals attraction, which strongly depends on the distance, and even a slight increase in the globule’s volume is enough to reduce it greatly. Generally, all of this is like the melting of a crystal, where a slight increase in volume reduces van der Waals interactions and liberates the motion of the molecules.

Unlike common polymer globules, the protein chain covered with different side chains cannot unfold by gradual swelling, because these side chains cannot change their positions independently; the rigid protein chain controls the positions of many side chains sitting at it, and this entire “forest” of side chains has to move as a whole.

Before the discovery of the MG state, protein denaturation was considered as complete unfolding of the protein structure; that is, as the transition to the coil. After this discovery [46,47,48], it became clear that the denatured protein can be either dense or loose, depending on the solvent’s strength and the hydrophobicity of the protein chain [2,45,46,47,48,49].

The pores in the MG (i.e., the vacant space necessary for side chain movements) are usually “wet”; that is, they are occupied by the solvent [45,50] because a water molecule inside the protein is still better than the vacuum. Experimentally, the “wetness” of the MG is proven by the absence of a visible increase in the protein partial volume [30] after denaturation of almost any type. A “dry” state of the pores is thermodynamically less stable [51,52], yet it is observed in some kinetic processes [26,51].

The MG compactness is maintained by residual hydrophobic interactions that are at least three times weaker than those within the native protein [31]; the fact that even these residual contacts are absent for some side chains emphasizes the heterogeneity of MG [9].

## 3. Cellular Events Causing Changes in Protein Structure Stability and Leading to the Transition to the MG

### 3.1. Post-Translational Modifications

In terms of protein structure, post-translational modifications can be equated to mutations. Most of them occur on the surface of a protein globule and do not have a significant effect on the protein structure. The result of some other modifications is the loss of the dense packing and transition to the MG-like state. Modifications of certain protein activities are known to require a change in ambient conditions [52] or local unfolding of the sites of the modifications [53,54,55], or (sometimes) partial denaturation [56], i.e., transition to a “softer” state.

For example, the tumor suppressor protein p53 is susceptible to a variety of modifications that change its functions in response to cellular stress, including acetylation, methylation, phosphorylation, and ubiquitination [56]. Specifically, p53 is the prime example of a protein whose acetylation requires partial denaturation. As another example: the peptidyl prolyl isomerase Pin1 can be modified by phosphorylation, ubiquitination, sumoylation, or oxidation, depending on the function that it will perform in the cell [57].

#### 3.1.1. Acetylation

Acetylation (mainly at Gln or Lys) is a reversible post-translational protein modification crucial for the regulation of gene expression [53,56,58]. It mainly affects large macromolecular complexes, such as chromatin remodeling, cell cycle, splicing, nuclear transport, actin nucleation, and others [53,54,57,59,60]. For histones, acetylation is critical because it triggers DNA transcription [61,62]. Moreover, acetylation can reduce interactions dependent on phosphorylation [61]. The acetylated N-terminus of a protein chain can create a specific signal for chain degradation [63].

Acyl groups that recognize elements of protein–protein interactions can vary from simple acetate to modified long-chain fatty acids required for the interaction with membranes and affecting signal transduction. They attach to various amino acids (Lys, Cys, and Ser/Thr), which can change the hydrophobicity of a protein and modify its functions. Myristoyl- and palmitate-induced modifications of Cys residues increase the protein affinity for membranes [64]. Thus, acylation is one of the key regulators of cellular pathways.

#### 3.1.2. Phosphorylation

Phosphorylation is the main mode of external signal transduction. ATP was shown to induce a conformational transition in proteins [65]. Signal transduction is associated with the modification of proteins that receive a signal from outside. These proteins interact with a large class of molecules known as adapter proteins and organizing centers (hubs) (SH2, SH3, 14-3-3, AKAPs), where the bound proteins undergo modification by phosphatases and kinases, and move within the cell and elsewhere [54,55,66].

#### 3.1.3. Ubiquitination

Ubiquitin (Ub) participates in many cellular events [67,68,69,70,71,72,73,74,75,76,77,78,79] including cell division, cell differentiation, signal transduction, movement of proteins within the cell, quality control, signaling, and endocytosis. Additionally, ubiquitin controls protein degradation and participates in DNA repair, endocytosis, autophagy, transcription, and immune system support. Ubiquitin binds to partners either as a single protein or in the form of Ub chains and interacts with Ub-receptors. The attachment of one to three Ub molecules affects the movement of proteins within the cell, and the addition of four or more (up to 16) Ub molecules directs the protein to degradation [69,70,71].

#### 3.1.4. Glycation

Glycation is another modification that affects the protein structure [80]. The addition of sugars manifests as early as during biosynthesis. Attachment of any type of sugar to in-protein residue results in the surface localization of this residue where it stays during the folding process. When the folding is completed, a specific enzyme removes the sugar molecule, but the labeled amino acid residue remains on the surface of the protein and participates in further events.

### 3.2. Protein–Protein Interactions

The interaction of protein ligands with protein receptors is a common event in the cell; one of the most studied interacting pairs is insulin and its receptor [81,82,83,84,85,86,87,88,89,90]. Using this system, it was demonstrated that during the interaction changes occur in the structures of both the hormone and the receptor. Both insulin and its receptor are water-soluble proteins. Independent experiments showed that in appropriate conditions, the denaturant-affected structures of these proteins undergo the N-to-MG transition. Surprisingly, when in a complex, the structures of both insulin and its receptor remain similar to their MG in solution. This is direct evidence that these proteins change their rigid structure to a softer one capable of interaction.

Similar changes were observed for the relaxin protein family related to insulin [91].

Importantly, during protein–protein interactions, a part of the surface that was in contact with water appears in a hydrophobic environment, which can cause rearrangement of the structure [81,82,83,84,85,86,87,88,89,90]. During the formation of multimeric and modular proteins, changes in their structures also result in more stable complexes [92,93].

### 3.3. Protein–Membrane Interactions

In 1988 we suggested that the membrane surface can influence the structure of proteins. Because of the proximity of a poorly (compared to water) polarized membrane, our experiments were performed at low dielectric permittivity and low pH. They evidenced the actual membrane impact on the protein structure (see reviews [25,94,95,96]). The study of apomyoglobin showed that in the presence of phospholipid vesicles at neutral pH, both its native and unfolded forms bind to the vesicle surface, showing properties similar to those of MG, i.e., both forms undergo conformational changes upon interaction with the membrane. For myoglobin the effect of membrane proximity is not so pronounced, but it is still significant. This influence is of decisive importance for the myoglobin-induced exchange of oxygen and carbon dioxide near the mitochondrial membrane.

### 3.4. Protein–Chaperone Interactions

Chaperones (specific assisting proteins) play a very important role in cell life. They bind to both native and unfolded proteins, providing their transition to the MG state and protecting them from random interactions, aggregation, and degradation [97,98,99,100,101,102].

Chaperones bind kinetic intermediates during protein folding, thereby shielding them from aggregation and side interactions. They ensure the correct folding of both monomeric proteins and oligomeric subunits, and the formation of oligomeric proteins [103]. Chaperones can transport newly synthesized proteins from the place of biosynthesis to the place of functioning. Their main role is to prevent non-specific interactions between proteins and keep them in a state competent for various processes in the cell. The binding of proteins to chaperones and their release consumes ATP energy [103]. Chaperones are also involved in protein quality control [72,104] and protein degradation [105].

### 3.5. Protein Interactions with Specific Adapter Proteins and Organizing Centers

Signal transduction within the cell requires precise work of the cellular regulatory system; to meet this requirement, specific protein domains interact with each other and different components of the cell. The modular nature of these domains allows their concurrent interaction with a variety of proteins [106,107]. They are known to recognize post-translational modifications [108,109,110,111,112,113], but because many of them are natively unfolded proteins they are not considered here.

However, among the adapters, there is at least one compact system that includes proteins of the 14-3-3 family [106,107,114,115,116]. The system received the strange name “14-3-3” due to the technique of isolation on chromatographic columns and testing by electrophoresis. For its main isoform (ζ), the crystal structure was deciphered back in 1995 [117,118]. It is a dimer consisting of two chains, each of which contains nine antiparallel helices; they form a horseshoe-like region between two structures where the substrate binds. There are nine isoforms of this protein, the combination of which allows the binding of multiple proteins. These isoforms recognize substrate proteins phosphorylated at Ser or Thr and can bind them by one or by two, depending on the cellular process. Such binding changes the conformation of the partner and brings the substrate proteins closer to one another, thus facilitating their interaction. It can also open an active site of the bound substrate, while other regions of this substrate can be shielded from currently unnecessary and even dangerous interactions with the main part of the protein [119,120,121].

The involvement of an adapter protein in the stabilization of the active site structure increases both the substrate binding and the yield of the reaction product, and this directly indicates the regulation of the catalytic activity of the bound substrate protein [117,122].

The structural basis for the interaction of 14-3-3 proteins with their substrates is described in [119,120,121]. As discovered, even disordered regions of the bound substrate protein acquire a well-structured shape upon interaction with a 14-3-3 protein [120,121,123]. Thus, it can be assumed that binding/release with 14-3-3 proteins can lead to the transition of the protein structure from rigid to soft and vice versa.

## 4. Functions of the Molten Protein Globule in the Cell

The development of new, more sensitive, and accurate research methods leads to the revision of some of the kinetic and equilibrium data, so the interpretation of these data may change [124,125]. For example, several proteins whose folding was previously believed to be a 2-state process were found to have compact folding intermediates with properties like those of MG [123,124]. This was observed for some cold shock proteins [126], the B1 domain of the G protein [127], single-chain monellin [128], RNase A [129], and the bacterial immune protein Im7 [130].

The experimental discovery of a “dry” MG as an unstable kinetic folding intermediate has presented some new data [26,52]; initially, the “dry” MG was predicted by Shakhnovich and Finkelstein [45,50] along with the much better known “wet” MG which is a stable folding intermediate. The presence of both the “dry” and “wet” MG was detected in monellin [26], the villin headpiece subdomain [131], *E. coli* DHFR [132], and RNase A [133].

Because some steps of enzymatic catalysis require structural flexibility, the biologically active conformational states other than fully folded structures may be more frequent than it was thought previously. For example, according to [134], the “non-native” state of acylphosphatase from *Sulfolobus solfataricus* shows enzymatic activity.

It was found that sometimes MGs occur outside the folding pathway in proteins with substituted amino acids, as exemplified by apoflavodoxin. Since MG are generally prone to aggregation, their presence outside the folding pathway increases the risk of protein accumulation, which can adversely affect the organism [134,135]. In 2003, Dobson comprehensively investigated how MG formation and aggregation can cause protein misfolding and aggregation, which results in numerous pathologies [136]. Later, he detected the other associations between the observed abnormalities in protein folding and diseases [136,137].

Surprisingly, the complex of human milk protein α-lactalbumin (αLA) with oleic acid (HAMLET) can kill cancer cells. In this complex, αLA is neither active nor “native” but preserves the MG state [138,139].

A similar complex with lysozyme also shows bactericidal activity, causing DNA fragmentation [139,140].

To date, ample evidence for the functional significance of many MG-like proteins has been reported [141]. This is true for the protein p53 [98], ferredoxin [142], alpha-mannosidase [143], melanogaster crammer [144], glycated serum albumin [80], and others. Some of them were reviewed by Bychkova et al., 2018 [10]. Moreover, the functional significance of such protein states is shown for the monomeric form of chorismate mutase [145], dihydrofolate reductase [146], ubiquitin [147], periplasmic binding proteins [148], staphylococcal nuclease [148], and α-galactosidase (cicer α-galactosidase) [149] (of course, we do not claim that a functional MG-like state is mandatory for all proteins).

An interesting observation concerning the metmyoglobin (MetMb) structure was reported back in 1998, although its altering conformation was not the focus of this study. The conformation changed during the transition of non-active MetMb to its active form capable of O_2_ binding. The transition pathway included an intermediate allowing further relaxation of the protein to form active deoxyMb, as was observed by the changing Soret band. This transition was catalyzed by MetMb reductase [150], which suggests that enzymes interacted with proteins in an intermediate state facilitating the reaction.

## 5. Conclusions

Summing up this review, we can state the fact that in the cell, some active proteins can have both the rigid structure typical of enzymes and that of an MG. In one of his re-views, Oleg B. Ptitsyn put forward a hypothesis that there could be two types of the “native” protein state, hard and soft. Since the “native state of a protein” usually implies the state that allows protein functioning, this prediction is apparently true for many proteins that display activity when their structure is MG-like. The experimental data obtained for different proteins speak for the proposed hypothesis that the transition of a protein to the state of an MG in a cell is not something extremely rare and exceptional. At least for some proteins, the state of the MG is necessary for the performance of their functions. This situation suggests the need to consider the conformational state of the protein when studying its activity both in the cell and in vitro, especially when projecting research results from one condition to another. The cell is a storehouse for unexpected phenomena and discoveries. Hopefully, the development of new and more sensitive methods or the improvement of those already in use will lead to the discovery of novel proteins functioning in the MG state.

## Figures and Tables

**Table 1 molecules-27-04361-t001:** Differences in the properties of the MG state of proteins and IDP.

Impact Type and Conditions	MG State, Globular Proteins	IDP State, Natively Unfolded Proteins
Unfolding by strong denaturants	Global unfolding to the coil state	Usually, no further global unfolding, but disruption of local structures is possible
Heat effect	Decrease in secondary structure content	Structuring, heat resistance
Different behavior of normalized SAXS curves (Kratky plots) [38,39]	Bell-shaped curves with a pronounced maximum	Monotonic curve rising (no maximum)
H/D exchange [40,41]	Slightly elevated exchange as compared to the N state	The exchange is orders of magnitude higher than that for MG
Gel filtration, electrophoresis [42,43]	Increase in hydrodynamic volume by 20–50% as compared to the N state	Hydrodynamic volume is 400–600% larger when compared to the N state of globular proteins with the same molecular weight

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
