# Peer review of "The Molten Globule State of a Globular Protein in a Cell Is More or Less Frequent Case Rather than an Exception"

_molecules, 2022, doi:10.3390/molecules27144361_

Round 1

Reviewer 1 Report

Journal: Molecules

Manuscript ID: molecules-1767849

Type of manuscript: Review

Title: The existence of a molten globule state of a globular protein in a cell is the rule rather than the exception

In this review, the authors proposed that the molten globular state of globular proteins are a functionally associated form in the cells. However, I have strong concerns regarding this statement or hypothesis.

1, Although several proteins have been shown to exhibit MG-like state in protein complexes, it is difficult to say that it is a “rule” that MG state was functional in the cells. Oleg Ptitsyn has proposed that the native state of proteins may be hard or soft, which is reasonable since the diverse functions of proteins rely on appropriate conformational changes upon binding with different partners. However, I disagree with the hypothesis that the so-called soft native state is limited to the MG or MG-like state and taken this hypothesis as a “rule”, which will cause misleading of either protein folding or the MG state. I agree with that the MG state may be functional in the cells, but do not think it is a “rule”.

2, The title should be revised to weaken the statement since no enough or solid evidences supporting this statement.

3, In section 3, as the authors stated, most of the post-translational modifications occur on residues located at the surface, clefts or termini of a protein, especially charged or polar residues. Generally, these residues may play a functional but not a structural role in protein structural maintenance, stability or folding. Therefore, modifications of these residues may induce protein conformational changes, but will not be a general “rule” to induce the MG state. Actually, most of the descriptions in this section could be classified as conformational changes or the soft state used by Oleg Ptitsyn, but not the MG state.

4, The manuscript contains  many confusing phrases or statements. Some examples,

-random coil but not coil;

-line 40, “but some IDPs properties uniquely distinguish them”?

-Table 1, “and conditions y”?

-Table 1, many intrinsically disordered proteins possess local structures. Denaturants could disrupt these local structures. Furthermore, some IDPs have a transition from the disordered to partially folded state upon protein binding or modifications.

-Lines 48-63, it is unnecessary to repeat the subtitles in the Introduction.

-line 80, “Since most of the protein chain degrees of freedom relate”?

-the styles of the Journal names are inconsistent in the references.

Author Response

We are extremely grateful to the reviewer for his careful reading and constructive criticism. Please read our answers below and the corrected manuscript in the attachment.

1) Reviewer`s comment:

Although several proteins have been shown to exhibit MG-like state in protein complexes, it is difficult to say that it is a “rule” that MG state was functional in the cells. Oleg Ptitsyn has proposed that the native state of proteins may be hard or soft, which is reasonable since the diverse functions of proteins rely on appropriate conformational changes upon binding with different partners. However, I disagree with the hypothesis that the so-called soft native state is limited to the MG or MG-like state and taken this hypothesis as a “rule”, which will cause misleading of either protein folding or the MG state. I agree with that the MG state may be functional in the cells, but do not think it is a “rule”.

Response to the reviewer:

The authors are grateful to the reviewer for careful reading and constructive criticism. We have tried to make the wording of our provisions and conclusions more correct. The word “rule” is not used now. In our understanding of the MG state and the «soft native» state are largely synonymous. To acquire functional softness, the protein needs to lose the rigidity of the side groups dense packing, but retain its structure and compactness, which are the main signs of a molten globule state. Such a transition may not cover the whole protein, but only a part of it, which is interpreted as the presence of several thermodynamic domains capable of independent conformational transitions in it.

2) Reviewer`s comment:

The title should be revised to weaken the statement since no enough or solid evidences supporting this statement.

Response to the reviewer:

Of course, we should use softer formulations. The title was revised and replaced with «The molten globule state of a globular protein in a cell is more or less frequent case rather than an exception».

3) Reviewer`s comment:

In section 3, as the authors stated, most of the post-translational modifications occur on residues located at the surface, clefts or termini of a protein, especially charged or polar residues. Generally, these residues may play a functional but not a structural role in protein structural maintenance, stability or folding. Therefore, modifications of these residues may induce protein conformational changes, but will not be a general “rule” to induce the MG state. Actually, most of the descriptions in this section could be classified as conformational changes or the soft state used by Oleg Ptitsyn, but not the MG state.

Response to the reviewer:

Of course, not all posttranslational modifications of proteins lead to conformational changes in the structure of proteins. Even less lead to the transition of the protein to the MG state. However, the presence of such facts suggests that this transition is not something unique. This effect is not necessary for all proteins, but is one of the ways of normal regulation of activity for some of them.

4) Reviewer`s comment:

The manuscript contains  many confusing phrases or statements. Some examples…

Response to the reviewer:

we thank the reviewer for his attention to detail when reading our manuscript. We have corrected the text in the specified places.

Reviewer 2 Report

This review summarizes many examples of proteins in a cell having structures that differ from both the native and unfolded states. In this manuscript, the protein states having the structures different from the native and unfolded states are called the molten globule (MG) state. Structures of the MG state have been extensively studied previously, but the term “MG state” has been less used in recent years. This manuscript is an interesting attempt to review recent literatures on protein structure and dynamics from the perspective of the MG state. Therefore, this reviewer considers that this manuscript merits publication in Molecules. However, there are a number of typographical errors, which should be corrected before publication.

1. The abbreviation of "molten globule (MG)" initially appears in page 1. However, in the latter pages, "molten globule" is sometimes used instead of MG.

2. In line 9 of Table 1, it says " Yield is 4-6 times higher …". What does "yield" mean here?

3. Line 61-63 says SH2, SH3, 14-3-3 adapters and AKAPs will be described in this section. However, in the section of "Protein interactions with specific adapter proteins and organizing centers", only 14-3-3 protein is described, and SH2, SH3 and AKAPs are not mentioned.

4. Line 73 says "S-shaped transition". What kind of measurements were performed to monitor the S-shaped transition?

5. The styles of many references are not in line with the format of this journal. In particular, please check the following points for all references: abbreviations of journal names, and whether to capitalize the first letter of words in article titles. If possible, please use Greek characters for alpha, beta, and so on.

6. Typographical errors:

In Table 1:

- Line 1: Impact type and conditions y --> delete "y"

- Line 3: the  coil --> delete an extra space

- Line 6: mag-nitude --> delete a hyphen

- Line 9: [41,41] --> [41]

Lines 61-63: Protein interactions with specific adapter proteins (dimers, trimers and multimers: SH2, SH3, 14-3-3 adapters and organizing centers AKAPs (A-kinase anchor proteins (AKAPs). --> "Protein interactions with specific adapter proteins (dimers, trimers and multimers: SH2, SH3, 14-3-3 adapters) and organizing centers (A-kinase anchor proteins (AKAPs))" or "Protein interactions with specific adapter proteins and organizing centers"

Line 75: H-bond --> hydrogen bond

Line 91: whole, --> whole.

Line 102: [46, 25] --> [25, 46]

Line 106: [9]). --> [9].

Line 116: is liable a --> is liable to a

Line 143: [61, 49, 50] --> [49, 50, 61]

Line 209: zeta should be in Greek

Line 226: 14-3-3  proteins --> delete an extra space

Line 237: [47, 25] --> [25, 47]

Line 240: ,  E. coli --> delete an extra space

Line 240: E. coli should be in italic.

Line 245: Sulfococus solfataricus --> Sulfolobus solfataricus (not Sulfococus). Also, the species name should be in italic.

Line 249: [127,  128] --> delete an extra space

Lines 254, 255, and 263: alpha should be in Greek

Line 256: [131,  132] --> delete an extra space

Line 261: [75 ] --> delete an extra space

Line263: is  shown --> delete an extra space

Line 266: MetMb --> metmyoglobin (MetMb)

Line 269: O2 --> 2 should be in subscript

Author Response

We are extremely grateful to the reviewer for his careful reading and constructive criticism. Please read our answers below and the corrected manuscript in the attachment.

1) Reviewer`s comment:

The abbreviation of "molten globule (MG)" initially appears in page 1. However, in the latter pages, "molten globule" is sometimes used instead of MG.

Response to the reviewer:

We have corrected this inaccuracy. Now MG is used everywhere after the first mention.

2) Reviewer`s comment:

In line 9 of Table 1, it says " Yield is 4-6 times higher …". What does "yield" mean here?

Response to the reviewer:

This phrase has been replaced with a more precise one: «Hydrodynamic volume is 400-600% large in compared to the N-state of globular proteins with the same molecular weight»

3) Reviewer`s comment:

Line 61-63 says SH2, SH3, 14-3-3 adapters and AKAPs will be described in this section. However, in the section of "Protein interactions with specific adapter proteins and organizing centers", only 14-3-3 protein is described, and SH2, SH3 and AKAPs are not mentioned.

Response to the reviewer:

We have removed this mention to avoid unjustified expectations of readers.

4) Reviewer`s comment:

Line 73 says "S-shaped transition". What kind of measurements were performed to monitor the S-shaped transition?

Response to the reviewer:

We have added a clarifying phrase: «…S-shaped transition observed by optical methods such as CD and fluorescence».

5) Reviewer`s comment:

The styles of many references are not in line with the format of this journal. In particular, please check the following points for all references: abbreviations of journal names, and whether to capitalize the first letter of words in article titles. If possible, please use Greek characters for alpha, beta, and so on.

Response to the reviewer:

We have checked and corrected this inaccuracies.

5) Reviewer`s comment:

Typographical errors: …

Response to the reviewer:

We are grateful to the reviewer for his careful reading and comments. All these comments and proposed corrections have found their place in the revised text.

Round 2

Reviewer 1 Report

In the revised manuscript, the authors has made some improvements regarding writting and has revised the title. However, little is considered for my previous comments 1 and 3.

The major "conclusion" of this review is better to be considered as a "hypothesis" rather than a "conclusion" since litte direct evidence supporting such a "conclusion. Therefore, it is necessary to weaken most of the statements throughout the manuscript, esp. those in the Abstract and Conclusion.

Furthermore, the authors has picked up some studies to support the hypothesis. esp. in Section 3. It is necessary to consider the effect of modifications and bimolecule interactions fair-minded and not prejudiced. Otherwise, it is easily to cause misleading.

Author Response

Dear reviewer!

We are grateful to the reviewer for the reasonable criticism of our article. Due to it, have made changes to make the article better.

Perhaps some misunderstanding of the authors' point of view by the reviewer is due to the difference in the interpretation of the term molten globule. The authors mean by this concept a compact state of the protein, which has a well-developed secondary structure, but has lost the dense packing of the side groups (see section 2 of our article). It is the loss of dense packaging that gives the protein the softness, as we mention in the article. From this point of view, an equality sign can be put between the soft native state and the functional molten globule (see modifications in sections 2, 3).

  • The major "conclusion" of this review is better to be considered as a "hypothesis" rather than a "conclusion"…

The hypothesis was put forward by Ptitsyn many years ago. Now this is mentioned already in Abstract and Conclusions. This review has collected experimental evidence in favor of this hypothesis. Whether they are enough for unambiguous confirmation, the reader can decide for himself. From our point of view, the facts are more than enough to confirm that in some cases the functioning of the protein requires a transition to the MG state (as well as to other non-rigid, but compact states). Of course, we do not claim that this is mandatory for all proteins. But, taking into account all the experimental data, this cannot be only attributed to some rare exceptions (see modifications in section 4 and Conclusions).

  • It is necessary to consider the effect of modifications and biomolecule interactions fair-minded and not prejudiced. Otherwise, it is easily to cause misleading.

We agree with the statement that not all modifications lead to significant conformational changes, and in no case we are biased when considering all the modifications. However, the experimental facts show that in some cases, during a modification, the protein structure changes to a more flexible one, and this change has a functional significance (see modifications in section 4).

Kind regards,

Vitalii Balobanov,
